# The Use and Potential Benefits of N-Acetylcysteine in Non-Acetaminophen Acute Liver Failure: An Etiology-Based Review

**DOI:** 10.3390/biomedicines12030676

**Published:** 2024-03-18

**Authors:** Mihai Popescu, Angelica Bratu, Mihaela Agapie, Tudor Borjog, Mugurel Jafal, Romina-Marina Sima, Carmen Orban

**Affiliations:** 1Department of Anaesthesia and Intensive Care, “Carol Davila” University of Medicine and Pharmacy, 37 Dionisie Lupu Street, 020021 Bucharest, Romania; agapiemili@yahoo.com (M.A.); tudorborjog@gmail.com (T.B.); jafalmugurel@yahoo.com (M.J.); carmen.orban@umfcd.ro (C.O.); 2Department of Anaesthesia and Intensive Care, Bucharest University Emergency Hospital, 169 Independentei Street, 050098 Bucharest, Romania; angibratu@yahoo.com; 3Department of Obstetrics and Gynecology, “Carol Davila” University of Medicine and Pharmacy, 37 Dionisie Lupu Street, 020021 Bucharest, Romania; romina.sima@umfcd.ro

**Keywords:** N-acetylcysteine, acute liver failure management, non-acetaminophen acute liver failure, survival

## Abstract

Acute liver failure represents a life-threatening organ dysfunction with high mortality rates and an urgent need for liver transplantation. The etiology of the disease varies widely depending on various socio-economic factors and is represented mainly by paracetamol overdose and other drug-induced forms of liver dysfunction in the developed world and by viral hepatitis and mushroom poisoning in less developed countries. Current medical care constitutes either specific antidotes or supportive measures to ensure spontaneous recovery. Although it has been proven to have beneficial effects in paracetamol-induced liver failure, N-acetylcysteine is widely used for all forms of acute liver failure. Despite this, few well-designed studies have been conducted on the assessment of the potential benefits, dose regimens, or route of administration of N-acetylcysteine in non-acetaminophen liver failure. This review aims to summarize the current evidence behind the use of this drug in different forms of liver failure.

## 1. Introduction

Acute liver failure (ALF) is a life-threatening condition associated with a decreased chance of survival if appropriate treatment is not undertaken. In the past, mortality rates reached 80% of cases and were mostly determined by the progression of cerebral edema and multiple system organ failure [1,2]. However, the introduction of advanced medical care led to an increase in spontaneous remission of liver failure, and consequently, mortality rates have currently decreased to under 30% [3]. This improvement in the critical care management of ALF patients has addressed, on one hand, the advanced support of associated organ dysfunctions [4] and, on the other hand, either supporting the spontaneous recovery of liver function or bridging patients to liver transplantation. Currently, standard medical care is hard to define, as there is wide variation among different centers, as well as across continents, based on personal knowledge, experience, and the availability of different treatments including liver grafts for liver transplantation, but it is mostly composed of both medical treatment and extracorporeal support. Although liver support devices did not show promising results in the management of patients with acute-on-chronic liver failure, a trend towards their efficacy has been associated with their use as part of the advanced management of ALF patients [5,6]. Despite this significant improvement in extracorporeal support, medical treatment addressing specific liver dysfunction is still lagging, due to various factors. On one hand, the incidence of ALF is not that high to start with, and it has declined significantly in the last decade [7]. On the other hand, the term ALF defines a multitude of etiologies, each one having its own prognosis, natural history, and, to some extent, specific management.

## 2. Pharmacological Rationale behind the Use of N-Acetylcysteine in Acute Liver Failure

N-acetylcysteine (NAC) has been long used as an antioxidant agent, administered to patients experiencing a paracetamol overdose. Paracetamol-induced hepatotoxicity is the leading cause of ALF in many countries [8]. Its pathophysiology is centered around the accumulation of a highly active intermediate compound, N-acetyl-p-benzoquinone imine (NAPQI), which is normally metabolized by glutathione. After high doses of paracetamol are ingested, glutathione stores are depleted, and NAPQI builds up and is responsible for hepatocellular necrosis [9]. NAC can restore glutathione stores and, hence, acts as a specific antidote in paracetamol overdose. 

Different studies have focused on the possible mechanisms behind the potential beneficial effects of NAC in non-acetaminophen overdose. Harrison et al. [10] showed that NAC increases the activation of guanylate cyclase, and this may be responsible for the anti-inflammatory, antioxidant, inotropic, and vasodilating effects demonstrated to improve liver blood flow and oxygen delivery to vital organs [11]. Also, NAC not only increases glutathione levels, but also modulates other mechanisms of oxidative stress and has been suggested to reduce endoplasmic reticulum stress and to improve mitochondrial function, both of which protect against liver injury [12]. These effects were observed on cell cultures treated with tuberculostatic drugs [13], further augmenting the potential beneficial effects of the drug in hepatoprotection. 

The main pathophysiological mechanism behind liver injury and subsequent organ dysfunction in ALF is represented by a severe systemic inflammatory response [14]. A recent meta-analysis looked at the effects of NAC in rebalancing this immune response [15]. The authors demonstrated a significant decrease in malondialdehyde and homocysteine levels and noted a decrease in major pro-inflammatory cytokine levels, including interleukin-8, interleukin-6, and tumor necrosis factor-α (TNF-α). Interestingly, there was no significant change in C-reactive protein levels. Moreover, a study by Stravitz et al. [16], investigating the effects of various cytokines on the progression of ALF, established a correlation between interleukin-17 concentration and unfavorable outcomes, and the beneficial effects of NAC in decreasing interleukin-17 levels. 

However, the effects of NAC extend beyond rebalancing inflammation at the level of the injured liver. In their experimental study on azoxymethane-induced hepatotoxicity in mice, Bémeur et al. [17] demonstrated that NAC led to a reduction in the extent of hepatic injury and to a subsequent improvement in brain function, the normalization of brain and hepatic glutathione levels, and a decrease in plasma pro-inflammatory cytokines, which may be clinically associated with a remission of organ dysfunctions in ALF. Moreover, reactive oxygen species (ROS) are responsible for the progression of liver dysfunction after an acute injury. In normal conditions, the liver cells have a complex defense system against ROS, including the expression of superoxide dismutase, glutathione peroxidase, and peroxiredoxins [18]. In acute injury, damage- or pathogen-associated molecular pathways are activated by Toll-like receptors found in Kupffer cells and, in turn, activate a pro-inflammatory cascade that upregulates the production of ROS [19]. In this way, ROS are the main mediators of the toxic and inflammatory destruction of hepatocytes, by inducing mitochondrial dysfunction via increasing intracellular oxidant stress, ultimately leading to cell necrosis [20]. In an animal study, Sun et al. [21] demonstrated the potential benefits of NAC in limiting this pathophysiological mechanism of liver injury, by inhibiting ROS-mediated endoplasmic reticulum stress, and, thus, showed that it may have significant theoretical effects in inhibiting the ROS-mediated apoptosis pathway [22]. A summary of current evidence on the pharmacological benefits of NAC is shown in Figure 1. 

Many studies have focused on the potential benefits of NAC in other non-paracetamol causes of ALF and reached conflicting results. The aim of this review is to summarize the current evidence on the use of NAC in patients with non-paracetamol-related ALF concerning the most common etiologies. 

## 3. Acute Liver Failure Due to Mushroom Poisoning

The incidence of ALF due to mushroom poisoning varies by region. In the United States, approximately 133,700 cases of ALF due to the ingestion of different species of poisonous mushrooms are reported each year [23]. In Europe, the incidence is not currently known, but more than 90% of fatalities after mushroom ingestion have been attributed to death cap mushrooms (amatoxins) [24]. The incidence may be higher in Eastern Europe compared to the Western world, as communities living near forests still handpick and cook wild mushrooms [25]. Liver failure is determined by the amatoxins contained within the mushrooms as they cause liver necrosis secondary to cellular damage due to fragmentation and segregation of all nuclear components [26]. The management of Amanita phalloides poisoning is aimed at non-specific measures such as correction of fluid and electrolyte imbalances, supporting organ functions, prevention of toxin absorption by either gastric lavage or the administration of activated charcoal, elimination of absorbed amatoxins, and administration of potential antidotes [27].

The pathophysiology of Amanita phalloides poisoning is complex. The main mechanism has been attributed to the binding of amatoxins and subsequent inhibition of ribonucleic acid (RNA) polymerase II which leads to a decline in messenger RNA levels and a subsequent decrease in protein synthesis that is responsible for the destruction of hepatocytes [28,29]. Toxicological and experimental studies have also demonstrated the central role of inflammation in mushroom-associated hepatotoxicity. In vivo studies showed that amatoxin actions and hepatotoxicity are enhanced by TNF-α, and animals that underwent pre-treatment with anti-TNF antibodies had less liver injury [30]. Moreover, α-amanitin is responsible for an increase in superoxide dismutase activity and ROS generation [31,32]. These supplemental mechanisms of enhanced hepatotoxicity secondary to mushroom poisoning may justify the use of NAC in the treatment of patients to limit systemic and local inflammation and decrease ROS formation. 

Many single case reports have been published on the use of high-dose NAC in Amanita phalloides poisoning, but unfortunately, current evidence is supported by either small case series or retrospective studies, with no properly designed randomized control trials published to date. In a first case series on both pediatric and adult patients published in 1992 by Locatelli et al. [33], the authors applied a prolonged infusion regimen of 150 mg/kg NAC bolus followed by 50 mg/kg every four hours for 3 to 18 days and reported a mortality rate of 8% regardless of liver transplantation. In an experimental study published in 1993, Schneider et al. [34] administered 1.2 g/kg NAC to an amatoxin-poisoned mouse model but could not demonstrate a significant change in either survival or improvement of liver functional tests (hepatic enzyme elevation) compared to controls. However, this initial negative study did not suppress researchers’ enthusiasm. Shortly after that, two small case series from the United States [35] and Italy [36] reported a decreased rate of mortality: two out of ten patients in the American cohort and none out of the eleven patients in the Italian cohort.

The research group of Locatelli et al. [36] published the first contemporary big cohort of amanita-induced ALF patients. In their research, they applied “the Pavia Protocol” consisting of 150 mg/kg intravenous NAC followed by 300 mg/kg/day until 48 h after poisoning, alongside forced diuresis until negative urinary amanitin levels were obtained. Their results showed a non-worsening of liver functional tests and a low mortality rate of 2.5%. In a case series of 77 patients published by Ahishall et al. [37], the authors reported using 70 mg/kg three times daily for the first 3 to 5 days, alongside digestive decontamination, specific antidotes (both penicillin G and silybin), and hemofiltration. They noted an improvement in liver functional tests and low mortality.

The first comparative non-randomized trial was published by Akin et al. [38] in 2013. In their research, they analyzed 40 patients out of whom 24 received a total of 12 g/day NAC intravenously divided into four doses and 16 patients received standard medical therapy only (including penicillin G). Their results showed an improvement in mortality and liver functional tests (serum transaminases, lactate dehydrogenase, and international normalized ratio) in the NAC group starting 24 h after the therapy commenced and lasting for the duration of the two-week follow-up. Although these results seemed promising, the study had significant limitations in terms of non-randomization, failure to report the severity of ALF, and the unstandardized protocol for the administration of NAC regarding body weight adjustments and administration as a continuous infusion similar to previous studies. The drawback regarding not considering the severity of liver dysfunction is important, as demonstrated by a US study published by Karvellas et al. [39]. In their cohort, they reported using an initial intravenous dose of 140 mg/kg NAC, followed by 70 mg/kg for 17 h, and continued after that with a variable dose at the discretion of the clinician. The main reported result was a large difference in mortality among patients with ALF compared to acute liver injury, both defined according to the United States Acute Liver Failure Study Group criteria.

A cohort study from the United States has recently been published [40]. The authors included 61 patients with mushroom poisoning and reported NAC as the only treatment option that was associated with decreased mortality. However, the authors acknowledge the possibility of a treatment bias in the sickest of patients. Finally, a recent systematic review [41] looked at 877 unique cases of mushroom poisoning from 133 publications. Due to the low number of cases reporting on the use of NAC alone, the outcome could not be evaluated. However, the addition of NAC to silybin was associated with higher survival rates compared to supportive care alone. 

In conclusion, no definite benefits of NAC use in patients with mushroom poisoning can be drawn. This is partially due to the lack of high-quality studies and partially because currently used doses are not individualized for this etiology, but rather imported from paracetamol-induced ALF protocols. On the other hand, many of the published studies look not only at severe ALF patients but a combination of ALF, acute liver injury, and patients with only mild symptoms. However, because virtually no significant side effects have been reported due to the use of NAC in this patient population, it can be administered as part of standard medical care, especially in regions where more advanced treatment options are lacking until further research elucidates its potential benefit. 

## 4. ALF Due to Viral Hepatitis

The incidence of viral-induced ALF has changed significantly over the last decade. On one hand, the incidence of hepatitis virus A (VHA)- or B (VHB)-induced ALF has steadily declined due to the increase in immunization but remains a problem in developing countries. On the other hand, hepatitis E virus is the most common etiology of viral hepatitis ALF due to its geographic spread in low-income countries [42]. However, many other hepatotropic viruses like herpes simplex virus/human herpesvirus, cytomegalovirus, Epstein–Barr virus, and parvovirus B19 have been linked to ALF [43]. Recently, the severe acute respiratory syndrome coronavirus 2 (SARS-CoV-2) virus has been associated with a high incidence of acute liver injury cases. Despite this disease being mild in the majority of cases, up to 6.4% of patients can progress to more severe forms [44]. 

Studies have demonstrated that besides having a direct hepatotoxic effect, most hepatitis viruses are associated with increased levels of interleukin-1β, TNF-α, interleukin-6, and interleukin-I receptor antagonists that are directly proportional to the severity of liver disease [45]. Moreover, viral infections determine an increase in oxidative stress and ROS formation that ultimately leads to cell death [46]. However, different mechanisms have been proposed depending on the type of viral infections, ranging from an imbalance in redox homeostasis with increased ROS production, oxidative stress, and decreased antioxidant enzymes [47] to decreased glutathione peroxidase and glutathione reductase levels [48]. Hence, the use of NAC may be beneficial in acute viral infections in both rebalancing the immune system and decreasing oxidative stress. 

There is a long tradition of using NAC in patients with acute-on-chronic liver failure and viral liver cirrhosis, with different studies showing potential benefits [49,50]. However, due to the difference in the natural history of the two forms of chronic liver failure and ALF, NAC has been less studied in patients with ALF. 

Several studies focused on the use of NAC in children with VHA-induced ALF. In one study [51], the authors observed 72 patients with VHA infection, of whom 12 had ALF and received NAC at a dose of 100 mg/kg orally every 4 h for the first 16 h, followed by the same dose at intervals of 6–8 h as long as required for normalization of laboratory tests. The authors reported a good tolerance, no adverse events, and a satisfactory clinical course. In the second study [52], the authors looked at 40 children with fulminant hepatic failure secondary to VHA infection. All patients received an NAC loading dose of 150 mg/kg over one hour, followed by 50 mg/kg for 4 h, and then a continuous infusion of 100 mg/kg over the next 16 h. Although they reported a high mortality of 38%, NAC was associated with an improvement in liver functional tests. Parkas et al. [53] divided 32 patients aged 5 to 13 into two groups based on whether they received NAC therapy at a dose of 100 mg/kg/day as a continuous infusion or not. They observed a decreased length of hospital stay and a trend toward improved survival in the treatment group. 

In adult patients, Mumtaz et al. [54] conducted a prospective non-randomized trial in patients with ALF. The majority of included patients, 73 out of 91 cases, had viral-induced ALF. Of these, 43 patients received NAC at a dose of 140 mg/kg, followed by 70 mg/kg for a total of 17 doses at 4-h intervals, and 30 patients were included in the control group. Although the results were not reported based on the etiology of ALF, the authors showed that the administration of NAC was associated with a higher survival rate. Guduez et al. [55] randomized 41 patients with VHA- or VHB-induced ALF to receive a low dose of 200 mg oral NAC three times daily or placebo. Their results showed no difference in the time necessary for the normalization of liver functional tests or a shorter duration of hospitalization. 

As previously mentioned, other non-hepatic viruses have been linked to the development of ALF. Of these, dengue fever is a common mosquito-borne viral infection, and many single cases and case series have been published on the use of NAC in patients with severe viral infections. In one case series, Dissanayake et al. [56] retrospectively analyzed 30 patients with dengue fever-associated severe hepatitis. In their protocol, they administered a continuous infusion of 100 mg/h NAC for 3 to 5 days and observed a faster and more significant recovery of liver enzymes following the administration of NAC. 

In conclusion, there are many case series on the potential benefits in terms of faster improvement of liver functional tests and decreased length of hospital stay of NAC use in children, especially those with severe VHA infections. However, most of this information comes from low-quality non-randomized studies, so larger, well-designed studies are required. Due to the low number of adults who develop severe acute viral hepatitis, no definitive conclusion can be drawn based on current research. 

## 5. Acute Liver Failure Due to Non-Paracetamol Drug-Induced Liver Injury

Due to the decline in acute viral hepatitis-induced ALF, drug-induced liver injury (DILI) represents one of the most common causes of ALF in developed countries [57]. Almost all cases of DILI are rather idiosyncratic than intrinsic because they affect only a small proportion of individuals treated with these drugs. Many antibiotics, antivirals, and non-steroid anti-inflammatory drugs have been linked to the development of DILI [58,59], and many studies have been published on the use of NAC in patients with DILI.

The pathophysiology of DILI is complex, with each drug class having its specific mechanism of hepatotoxicity. Nevertheless, the imbalance of redox activity is a crucial step in DILI as most hepatotoxic drugs determine a build-up of ROS and induce oxidative stress by different mechanisms such as an increase in intracellular oxidants and depletion of antioxidants that finally lead to mitochondrial dysfunction [60,61]. In the case of most drugs, this increase in oxidative stress has been well documented, while in a small proportion of drugs, it remains debatable [62]. A joint study of the Drug-Induced Liver Injury Network and the Acute Liver Failure Study Group has found that different patterns of cytokines predict the outcome of DILI, although none were reflective of etiology [63]. Based on animal models, it appears that TNFα and interferon-γ act synergistically to promote cell death and inhibit hepatocyte regeneration [64]. Based on these mechanisms, studies have focused on the potential role of NAC as an antioxidant drug outside its use in acetaminophen overdose. 

Tuberculostatic therapy represents one of the most common causes of DILI in the developing world with a high incidence of tuberculosis. In one study, Shadi et al. [65], looked at the prophylactic effects of NAC in the development of liver injury after a four-drug regimen for tuberculosis. They randomized 60 patients to receive NAC at a dose of 600 mg orally twice daily for two weeks or a placebo. The addition of NAC was associated with a significantly lower increase in serum transaminase, and none of the patients developed hepatotoxicity compared to an incidence of 37.5% in the placebo group. These results may be explained by an animal study conducted by Attri et al. [66] about the effects of NAC on oxidative stress. They showed that the oxidative stress secondary to isoniazid and rifampicin administration was caused by a significant decline in glutathione and related thiols and that the administration of 100 mg/kg/day NAC was associated with decreased oxidative stress. Moosa et al. [67] conducted a double-blind randomized control trial on the effect of NAC at a dose of 150 mg/kg over 1 h followed by 50 mg/kg over 4 h and 100 mg/kg over 16 h in 53 patients with anti-tuberculosis-drug-induced liver injury. Although they did not observe a significantly faster decrease in serum transaminase, they did note a significantly shorter length of hospital stay. 

Other case series looked at the potential effects of NAC in non-tuberculostatic DILI. In a case–control study [68], the authors assessed 102 pediatric patients with chemotherapy-induced hepatotoxicity. Of these, 70 patients received NAC at a dose of 3 μg/kg intravenously in a 24-h infusion and had a faster improvement in liver functional tests compared with the control group. Other small case series or case reports noted potential benefits of NAC in DILI secondary to amiodarone [69], antibiotics [70], or dietary supplements [71]. 

Three meta-analyses have been published on NAC administration in DILI. In the first one, Chughlay et al. [72] demonstrated that NAC was associated with improved transplant-free survival in patients with non-paracetamol DILI but did not affect overall survival. In the second systematic review of 11 studies, Cabrera et al. [73] failed to perform an adequate meta-analysis due to the high risk of bias and differences in methodology between studies. However, they observed a trend towards improvement of transplant-free survival without reaching a definitive conclusion on overall survival. Also, they noted that in prevention studies NAC demonstrated a possible hepatoprotective effect and favorable safety profile. In the third meta-analysis, Shrestha et al. [74] included 11 studies that analyzed 1107 patients: 565 in the NAC group and 552 in the control group. Their results showed that the use of NAC was associated with a 53% decrease in mortality compared to standard of care and a reduction by 6.5 days in the mean length of hospital stay. They also reported a decrease in the incidence of hepatic encephalopathy by 59%, but a higher risk for nausea and vomiting and the need for mechanical ventilation. 

In conclusion, different studies have shown a positive trend in using NAC for the prevention of DILI, especially in patients taking drugs with a high risk for liver injury. Regarding the use of NAC for the treatment of established DILI, current studies and meta-analyses favor its use compared to standard of care for decreasing mortality and length of hospital stay. However, these results should be interpreted with caution as many of the published studies have a high risk of bias and did not focus on a specific drug class. More studies are needed to estimate the impact of NAC on drug-specific DILI stratified by the severity of liver dysfunction. 

## 6. Pregnancy-Related Acute Liver Failure

Pregnancy-related ALF is associated with significant maternal and fetal mortality [75]. The most common etiologies are represented by hemolysis, elevated liver enzymes, low platelets (HELLP) syndrome and acute fatty liver of pregnancy (AFLP). However, a recent report by the American Acute Liver Failure Study Group noted that acetaminophen toxicity was responsible for most cases and that the classic forms of HELLP and AFLP account for only a minority of ALF cases [76]. Because of this, there is only a limited number of studies investigating the use of NAC in pregnancy-related ALF, most of which are single case reports [77]. 

Classically, HELLP syndrome has been associated with the presence of hypertension, severe inflammation, increased oxidative stress, and endothelial activation. Management options have been focused on limiting these pathophysiological mechanisms [78]. Most inflammatory cytokines, like interleukin-6 and TNF-α, are elevated in both women with HELLP syndrome [79] and their newborns [80]. These cytokines are responsible for the imbalance between lipid peroxidation and antioxidant defense mechanisms and, in turn, lead to endothelial activation and free-radical-mediated cell damage [81]. Thus, theoretically, NAC may have a positive role in both the prevention and limitation of the effects of liver damage in pregnancy-related ALF. 

In one randomized controlled trial, the authors [82] investigated 60 patients with severe pre-eclampsia requiring termination of pregnancy and divided them into two groups: the first group received 70 mg silymarin as a single dose, and the second group received 600 mg NAC at 0, 12 and 24 h after delivery. Their results showed an improvement in liver functional tests in both study groups, with a trend towards a better outcome in patients receiving silymarin. In another study, Roes et al. [83] randomized 38 patients with early-onset preeclampsia and/or HELLP to receive either 1800 mg NAC every 8 h or placebo tablets. They reported that the use of oral NAC did not improve maternal and neonatal outcomes in terms of organ dysfunction and intensive care unit length of stay. 

In conclusion, current evidence does not support the use of NAC for pregnancy-related ALF. However, there is limited evidence of its use due to the low number of published trials, and it should not be yet disregarded until future studies further assess potential benefits. 

## 7. Pooled Analyses of Acute Liver Failure Etiologies

As previously mentioned, the incidence of non-acetaminophen ALF is low in the general population and many well-designed studies have chosen to undertake a non-etiological assay on the potential benefits of NAC as they included patients regardless of the underlying cause of ALF as well as patients with indeterminate etiology. 

In a prospective double-blind trial [84], 173 patients with non-paracetamol ALF were randomized to receive either placebo or NAC at a loading dose of 150 mg/kg/h for 1 h, followed by 12.5 mg/kg/h for 4 h and 6.25 mg/kg/h for 67 h. The most common etiologies for ALF were DILI, autoimmune hepatitis, hepatitis B virus, and indeterminate causes. The authors stratified their results based on the severity of hepatic encephalopathy at the time of inclusion and demonstrated that NAC-treated patients with mild hepatic encephalopathy (grade I and II) had better survival compared to controls. This observation suggests that NAC may be beneficial in patients with non-severe neurologic dysfunction and that it should be administered at the earliest time possible before severe encephalopathy develops. They demonstrated improved transplant-free survival with the use of NAC, although this was noted only in patients with mild hepatic encephalopathy, but otherwise reported no difference in overall survival. The authors noted a non-significant trend towards a lower incidence of liver transplantation in the treatment group and a significant increase in the incidence of nausea and vomiting as the main side effect. In a case–control study [54], the authors compared 47 patients prospectively receiving 140 mg/kg oral NAC, followed by 70 mg/kg at 4-h intervals for a total of 17 doses with a historical cohort of 44 patients. The majority of patients had acute viral hepatitis or DILI. Patients receiving NAC were younger and had higher bilirubin levels and significantly lower mortality rates. Also, the authors looked at the effects of NAC on neurologic function. Although there was no difference in the severity of hepatic encephalopathy between the two groups, NAC-treated patients had a higher incidence of signs of raised intracranial pressure (dilated pupils and seizure activity) and required more measures to decrease intracranial pressure (use of mannitol, hyperventilation, and cooling blankets) compared to patients in the control group. This signal towards a worse neurologic outcome associated with the use of NAC was not observed in other trials but needs further investigation. In another multicenter study, Darweesh et al. [85] randomized 155 patients (most of them with viral hepatitis followed by DILI) to receive either placebo or NAC 150 mg/kg for half an hour, followed by 70 mg/kg for 4 h and then 70 mg/kg for 16 h and 150 mg/kg/day continuously until two consecutive normal international normalized ratios were obtained. They observed a decrease in mortality, need for transplantation, length of hospital stay, and incidence of hepatic encephalopathy and bleeding, as well as an improvement in liver functional tests in the treatment group. In a randomized control trial, Nabi et al. [86] compared 40 ALF patients treated with NAC to 40 controls. The main etiologies were represented by viral hepatitis and DILI. The authors stratified patients based on the severity of their hepatic encephalopathy at the time of presentation. Although not statistically significant, patients in the treatment group had a more severe encephalopathy. Their results showed that patients in the NAC group had better survival, although they had worse prognostic factors, and more patients had grade IV encephalopathy compared to controls. Also, they reported a better survival of 100% in DILI patients treated with NAC. 

Two meta-analyses were performed on the use of NAC in non-acetaminophen ALF. In the first one, published in 2015, Hu et al. [87] included four studies and a total of 616 patients: 331 patients in the NAC treatment group (administered either orally or intravenously) and 285 patients in the control group. Although there was no difference in overall survival, a significant difference was noted in both incidence of spontaneous recovery and post-transplantation survival. The most frequently noted side effects were nausea, vomiting, diarrhea, and constipation, but a link to NAC therapy could not be established. In the second meta-analysis, Walayat et al. [88] included seven prospective and retrospective studies in both adult and pediatric patients with non-acetaminophen ALF. The most common etiologies for ALF were viral hepatitis, DILI, and indeterminate causes. There was no restriction on the dose or route of NAC administration. Their results showed a significant decrease in overall, transplant-free, and post-transplant survival rates and noted a shorter length of hospital stay associated with NAC use. 

A summary of current evidence regarding different etiologies of non-acetaminophen ALF is presented in Table 1. 

## 8. Conclusions

Current evidence supports the use of NAC in patients with ALF of different etiologies as it may be associated with potential benefits in survival and remission of liver injury due to its effects on both pro-inflammatory cytokines and reactive oxygen species. However, there is a huge variability in the time of initiation, route of administration, dose, and duration of therapy. Because of heterogeneity and the lack of high-evidence randomized controlled trials, we were not able to perform an etiology-based meta-analysis on the use of NAC in ALF. Moreover, there is also huge variability regarding patients’ age, as some etiologies, like acute viral hepatitis, are more frequent in the pediatric population and others, like DILI, are more common in elderly patients. Hence, age and its relation to liver regenerative capabilities and the extent of inflammation may have a significant impact on patients’ outcomes that should further be assessed. As the incidence of non-paracetamol ALF is low in the general population, there is an urgent need for international collaboration to perform multicenter, well-conducted trials that take into account the aforementioned limitations of current research. 

In conclusion, if we disregard the etiology of non-paracetamol ALF, current research suggests the potential benefits of NAC in increasing transplant-free survival, but definitive evidence is still lacking regarding overall survival. Current practice in most centers involves the administration of high-dose intravenous NAC starting with 150 mg/kg for the first hour and decremental doses afterward in patients with non-acetaminophen ALF, but more studies are needed to properly investigate both the dose and the route of administration. Regarding the use of NAC in an etiology-specific population, current evidence does not support either the use or the disuse of NAC in different etiologies. However, since most studies reported either no or only mild side effects, like nausea and vomiting that could not be attributed to the treatment itself, we consider that NAC can be used as part of standard medical care until future research establishes specific indications in various etiologies.

## Figures and Tables

**Figure 1 biomedicines-12-00676-f001:**
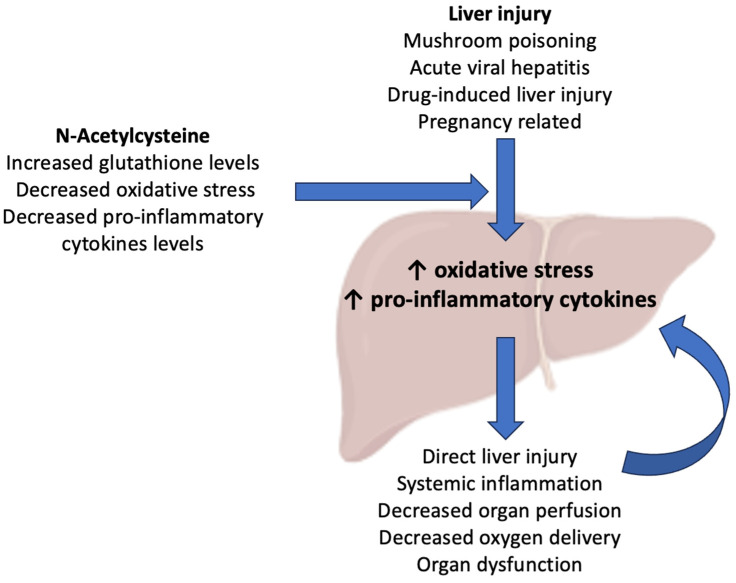
Potential mechanism of action for N-acetylcysteine (NAC) in patients with non-acetaminophen acute liver failure.

**Table 1 biomedicines-12-00676-t001:** Current evidence on the use of N-acetylcysteine in patients with non-paracetamol acute liver failure.

Author, Year	Type of Study	Number of Cases	Age of Participants (Years)	Dose of NAC	Co-Treatment/ Comparator	Observed Effects	Mortality
Mushroom poisoning (*Amanita phalloides*)							
Locatelli C. et al., 1992 [27]	Case series	73	2–84	150 mg/kg bolus, followed by 50 mg/kg at 4 h intervals for 3–18 days	Forced diuresis	- low mortality regardless of LT	8%
Schneider S. et al., 1992 [33]	Experimental animal model	NA	NA	1.2 g/kg	None	- no benefits on survival - no improvement in LFT	NA
Locatelli C. et al., 2010 [36]	Cohort of patients	157	Mean: 52	150 mg/kg followed by 300 mg/kg/day until 48 h	Forced diuresis, activated charcoal	- no progression of severity of liver dysfunction	2.5%
Ahishali E. et al., 2012 [37]	Cohort of patients	77	Mean: 42	210 mg/kg/day for 3–5 days	Activated charcoal, Penicillin G, Sylibin, hemofiltration	- improvement in LFT - low mortality	2.6%
Akin A. et al., 2013 [38]	Case–control	40 patients: 24 in the NAC group vs. 16 in the control group	NAC: mean 32 Control: mean 34	12 g/day in four divided doses	Standard medical care	- decreased mortality - improved LFT from 24 h to 2 weeks in the NAC group	4.4% in the NAC group vs. 18.7% in the control group
Tan J. et al., 2022 [41]	Review of 133 studies	877 cases	NA	variable	NR	- improved survival of patients receiving NAC and silybin	NR
Viral hepatitis							
Parkas A. et al., 2016 [53]	Cohort of patients	32 patients: 12 receiving NAC and 12 control	NAC: mean 7.2 Control: mean 7.3	100 mg/kg/day	Standard medical care	- decreased hospital LoS - non-significant decrease in mortality	44% in NAC group vs. 69% in control group
Gunduz H. et al., 2003 [55]	Randomized control trial	41 patients with VHA or VHB ALF	15–52	200 mg orally, three times per day	Standard medical care	- no improvement in LFT - no decrease in hospital LoS	NR
Dissanayake D. et al., 2021 [56]	Case series	30 patients with dengue fever and severe hepatitis	Mean: 49.9	100 mg/h for 3–5 days	Standard medical care	- faster recovery of liver enzymes after NAC administration	3.3%
Drug-induced liver injury							
Moosa M. et al., 2020 [67]	Randomized control trial	53 patients with tuberculostatic-induced liver injury	NAC: mean 37 Placebo: mean 38	150 mg/kg over 1 h, 50 mg/kg over 4 h, and 100 mg/kg over 16 h	NR	- no faster improvement in LFT - decreased hospital LoS	14%
Eroglu N. et al., 2020 [68]	Case–control	102 pediatric patients with chemotherapy-induced hepatotoxicity: 70 received NAC	2–17 in both groups	3 μg/kg in a 24 h infusion	NR	- faster improvement of LFT	NR
Chughlay M. et al., 2016 [72]	Meta-analysis	45 patients: 19 in NAC group vs. 27 controls	NR	150 mg/kg/h over 1 h, then 12.5 mg/kg/h for 4 h, then 6.25 mg/kg/h for 67 h	NR	- improved transplant-free survival - no improvement in overall survival	21% in NAC group vs. 35% in controls
Sanabria-Cabrera J. et al., 2022 [73]	Systematic review	11 studies	NR (≥18)	Varying in different studies	NR	- improved transplant-free survival	NR
Shrestha D. et al., 2021 [74]	Meta-analysis	11 studies: 565 patients in the NAC group vs. 552 in the control group	NR	Varying in different studies	standard of care	- decreased mortality - shorter hospital LoS - improved HE	23.7% in NAC group vs. 35.1% in control group
Pregnancy-related ALF							
Shabani S. et al., 2021 [82]	Randomized control trial	60 patients with severe pre-eclampsia	Mean: 26	600 mg at 0, 12, and 24 h after delivery	70 mg of silymarin	- improvement of LFT in both groups	NR
Roes E. et al., 2006 [83]	Randomized control trial	38 patients with severe pre-eclampsia and/or HELLP	NAC: 22–34 Control: 23–40	1800 mg every 8h	placebo	- no improvement in organ dysfunction or LoS	NR
Pooled analysis (all etiologies)							
Lee W. et al., 2009 [84]	Randomized control trial	81 patients in the NAC group vs. 92 patients in the placebo group	NAC: 17–69 Control: 18–71	150 mg/kg/h for 1 h, followed by 12.5 mg/kg/h for 4 h and 6.25 mg/kg/h for 67 h	placebo	- increased transplant-free survival (in mild HE patients) - no difference in overall survival	30% in NAC group vs. 34% in control group
Mumtaz K. et al., 2009 [54]	Prospective study	47 patients in NAC group vs. 44 patients in historical cohort	NAC: 27 Control: 37	140 mg/kg orally, followed by 70 mg/kg at 4 h intervals for a total of 17 doses	Standard of care	- improved survival - worse neurological outcome	53.2% in NAC group vs. 72.7% in historical cohort
Darweesh S. et al., 2017 [85]	Randomized control trial	85 patients in the NAC group vs. 70 patients in the control group	NAC: 33 Placebo: 34	150 mg/kg for 1/2 h, followed by 70 mg/kg for 4 h, then 70 mg/kg for 16 h, and 150 mg/kg/day afterward	placebo	- improved overall survival - decreased need for LT - improved LFT - decreased incidence of HE	3.3% in treatment group vs. 23.3% in control group
Hu J. et al., 2015 [87]	Meta-analysis	4 studies: 331 patients in the NAC group vs. 285 patients in the control group	NR (adults and children)	orally or intravenously, in different doses	NR	- increased transplant-free survival - increased survival after LT - no difference in overall survival	29.0% in treatment group vs. 33.0% in control group
Walayat S. et al., 2021 [88]	Meta-analysis	7 studies: 472 patients in the NAC group vs. 411 in the control group	NAC: 21 Control: 26	orally or intravenously, in different doses	NR	- increased overall, transplant-free, and post-LT survival - shorter hospital LoS	23.7% in treatment group vs. 35.7% in control group

Abbreviations: NAC—N-acetylcysteine; LT—liver transplantation; h—hour; NA—not applicable; LFT—liver functional test; NR—not recorded; LoS—length of stay; VHA—viral hepatitis A; VHB—viral hepatitis B; ALF—acute liver failure; HE—hepatic encephalopathy.

## Data Availability

Not applicable.

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
