# Peer review of "The Use and Potential Benefits of N-Acetylcysteine in Non-Acetaminophen Acute Liver Failure: An Etiology-Based Review"

_biomedicines, 2024, doi:10.3390/biomedicines12030676_

Round 1

Reviewer 1 Report

Comments and Suggestions for Authors N-Acetylcysteine should be abbreviated as NAC- instead of NACC. The text included in the Figure 1 is not clear, please re-design the figure with clear letters. To complete at row 104 and below Amanita phalloides (there are numerous species of mushrooms within the Amanita genus, but in the mansucript is about A. phalloides). To replace NCC with NAC at row 283. In Table 1, to replace ‘Anti-tuberculostatic’…either is tuberculostatic, or is Anti-tuberculosis drug. In section 8., Conclusion, please include the doses of NAC recommended (in mg/kg).

Author Response

Dear reviewers,

Thank you very much for both your time used in reading and reviewing our manuscript, as well as your input and suggestions made for the improvement of our paper which contributed a lot to the overall quality of our review. We hereby attach a point-by-point response to your comments and suggestions. All the changes made to the manuscript have been highlighted in yellow.

Reviewer 1.

  1. N-Acetylcysteine should be abbreviated as NAC- instead of NACC. Indeed, NAC is a more commonly used abbreviation for N-Acetylcysteine, and we have changed it throughout the manuscript.
  2. The text included in the Figure 1 is not clear, please re-design the figure with clear letters. We have redesigned Figure 1 for a better understanding of the text.
  3. To complete at row 104 and below Amanita phalloides (there are numerous species of mushrooms within the Amanita genus, but in the manuscript is about A. phalloides). The review is correct, although many mushroom species have been implicated in acute liver toxicity, most studies have focused on one specific Amanita – Amanita phalloides. We have changed this in the “mushroom poisoning section”.
  4. In Table 1, to replace ‘Anti-tuberculostatic’…either is tuberculostatic or is Anti-tuberculosis drug. We have replaced with “tuberculostatic” drugs.
  5. In section 8., Conclusion, please include the doses of NAC recommended (in mg/kg). We have included a new paragraph in the conclusion section of our manuscript, containing both future research directions, as well as the recommended dose of NAC in mg/kg.

Reviewer 2 Report

Comments and Suggestions for Authors

In the liver N-acetylcysteine (NAC) forms cysteine, which is the rate limiting amino acid in glutathione synthesis. Hepatotoxicity of high doses of paracetamol is the consequence of insufficient inactivation of the hepatotoxic metabolite N-acetyl-p-benzochinone imine by glutathione (GSH) conjugation because GSH formation is overwhelmed. Application of N-acetylcysteine (NAC) enhances GSH synthesis and by that provides sufficient GSH for inactivation of the hepatotoxic metabolite, explaining the protective effect of NAC in paracetamol intoxication. Moreover, GSH is an antioxidant which scavenges reactive oxygen species (ROS), which are formed during inflammation. With that, NAC may impair toxicity of ROS formed during inflammatory reactions. This may explain protective effects of NAC in diseases, which result in inflammatory reactions.

Based on the well-known protective effect of NAC in paracetamol poisoning the authors have searched the literature for protective effects of NAC applied in different hepatic diseases. Such information can easily be found in the literature and only warrants publication if the authors provide plausible arguments, why or why not NAC may be effective either by inactivating reactive metabolites of the specific hepatotoxic drug or by scavenging increased ROS formation in case of inflammation.

In case such information is provided the manuscript may become acceptable for publication.

Comments on the Quality of English Language

Mi

nor editing of English

Author Response

Dear reviewers,

Thank you very much for both your time used in reading and reviewing our manuscript, as well as your input and suggestions made for the improvement of our paper which contributed a lot to the overall quality of our review. We hereby attach a point-by-point response to your comments and suggestions. All the changes made to the manuscript have been highlighted in yellow.

Reviewer 2.

  1. Based on the well-known protective effect of NAC in paracetamol poisoning the authors have searched the literature for protective effects of NAC applied in different hepatic diseases. Such information can easily be found in the literature and only warrants publication if the authors provide plausible arguments, why or why not NAC may be effective either by inactivating reactive metabolites of the specific hepatotoxic drug or by scavenging increased ROS formation in case of inflammation. In case such information is provided the manuscript may become acceptable for publication. Thank you very much for your very important input. In trying to summarize current evidence for everyday clinicians, we forgot to justify the use of NAC in the case of ALF. This is of crucial importance as it offers both additional information on the pathophysiology of ALF and links it to the use of NAC, as well as offers a possibility to identify those patients who, based on current evidence, would benefit most from NAC administration (those with severe inflammation as assessed by inflammatory markers etc.) until future research and/or guidelines are available. We have updated our manuscript accordingly to your suggestion in two ways: first, we have updated the second chapter of the manuscript “Pharmacological rationale behind the use of N-Acetylcysteine in ALF” to include more inflammation of the effects of NAC on cytokines and ROS and linked it to the pathophysiology of ALF in general. Secondly, we have included in each chapter of ALF aetiology, some important information on the effects of specific aetiologies on the liver in terms of inflammation and ROS formation etc. and by NAC may have potential benefits in each case (or not at all, it should be avoided). Once more, thank you very much for this very important idea.  

Reviewer 3 Report

Comments and Suggestions for Authors

Review of the paper entitled „The use and potential benefits of N-acetylcysteine in non-acetaminophen acute liver failure: an aetiology-based review by Mihai Popescu, Angelica Bratu, Mihaela Agapie, Tudor Borjog, Mugurel Jafal, Romina Sima and Carmen Orban

 My comments

 The topic discussed by the Authors is interesting. However, the paper requires some improvement.

There is no information regarding the age of the patients. Treatment results may vary between age groups. The authors also did not provide the etiology of ALF. Moreover, the authors did not determine the benefits of using N-acetylcysteine depending on the degree of hepatic encephalopathy. Literature data indicate both the benefit of using NACC in advanced hepatic encephalopathy and the lack of benefit of using NACC in the treatment of stage III–IV hepatic encephalopathy [Nabi T, Nabi S, Rafiq N, Shah A. Role of N-acetylcysteine treatment in non-acetaminophen-induced acute liver failure: A prospective study. Saudi J Gastroenterol. 2017 May-Jun;23(3):169-175. doi: 10.4103/1319-3767.207711. PMID: 28611340; PMCID: PMC5470376; Mumtaz K, Azam Z, Hamid S, Abid S, Memon S, Ali Shah H, Jafri W. Role of N-acetylcysteine in adults with non-acetaminophen-induced acute liver failure in a center without the facility of liver transplantation. Hepatol Int. 2009 Dec;3(4):563-70. doi: 10.1007/s12072-009-9151-0. Epub 2009 Aug 29. PMID: 19727985; PMCID: PMC2790590; Lee WM, Hynan LS, Rossaro L, Fontana RJ, Stravitz RT, Larson AM, Davern TJ 2nd, Murray NG, McCashland T, Reisch JS, Robuck PR; Acute Liver Failure Study Group. Intravenous N-acetylcysteine improves transplant-free survival in early stage non-acetaminophen acute liver failure. Gastroenterology. 2009 Sep;137(3):856-64, 864.e1. doi: 10.1053/j.gastro.2009.06.006. Epub 2009 Jun 12. Erratum in: Gastroenterology. 2013 Sep;145(3):695. Dosage error in article text. PMID: 19524577; PMCID: PMC3189485].

 The quality of Figure 1 should be improved.

Author Response

Dear reviewers,

Thank you very much for both your time used in reading and reviewing our manuscript, as well as your input and suggestions made for the improvement of our paper which contributed a lot to the overall quality of our review. We hereby attach a point-by-point response to your comments and suggestions. All the changes made to the manuscript have been highlighted in yellow.

Reviewer 3.

  1. There is no information regarding the age of the patients. Treatment results may vary between age groups. Thank you very much for raising our awareness on this very important issue. Indeed, age may have a significant impact on the results. In order to correct this, we have included in table 1 a separate column containing the age of participants. We tried to include the information in the text as well, but we considered it made the manuscript very difficult to read and hence we opted to only mention if the study was conducted on a paediatric population. Moreover, due to the importance of this issue we have included this in our discussion section as one of the current limitations of current literature and future perspectives for research.
  2. The authors also did not provide the etiology of ALF. The review is structured based on aetiology and each main cause of ALF has its own chapter: “mushroom poisoning”, “acute hepatitis” etc. In case of different causes of ALF in a chapter e.g. “viral hepatitis”, we have included the cause of ALF (hepatitis A, Dengue fever) etc. Also, in table 1, there is a subsection for each cause of ALF. We have updated the “pooled analysis” section to include the cause of ALF in the studies discussed (we did this only in the manuscript as it would have made the table too hard to follow and unstructured).
  3. Moreover, the authors did not determine the benefits of using N-acetylcysteine depending on the degree of hepatic encephalopathy. Literature data indicate both the benefit of using NACC in advanced hepatic encephalopathy and the lack of benefit of using NACC in the treatment of stage III–IV hepatic encephalopathy. Thank you very much for this very interested observation. We have updated our reference so that all the studies mentioned in your recommendations were included. Although we cited most of them already, we failed to compare the effects of NAC on HE. Our manuscript have been updated so it now contains this crucial yet still debated (as many pro’s and con’s exist) information and has also been included in table 1.
  4. The quality of Figure 1 should be improved. We have redesigned Figure 1 for a better understanding of the text.

Round 2

Reviewer 1 Report

Comments and Suggestions for Authors

The manuscript was well improved, got a lot better after taking into account the reviewers' suggestions.

Reviewer 2 Report

Comments and Suggestions for Authors

Based on the reviewer’s comments the authors have significantly improved their manuscript so that I recommend acceptance for publication.

No further comments.

Comments on the Quality of English Language

Minor editing recommended

Reviewer 3 Report

Comments and Suggestions for Authors

The Authors revised their manuscript taking my comments into account.            I believe that in this version the paper can be published in Biomedicines journal.